# Upregulation of TLR4-Dependent ATP Production Is Critical for *Glaesserella parasuis* LPS-Mediated Inflammation

**DOI:** 10.3390/cells12050751

**Published:** 2023-02-26

**Authors:** Fei Liu, Yidan Gao, Jian Jiao, Yuyu Zhang, Jianda Li, Luogang Ding, Lin Zhang, Zhi Chen, Xiangbin Song, Guiwen Yang, Jiang Yu, Jiaqiang Wu

**Affiliations:** 1Shandong Key Laboratory of Animal Disease Control and Breeding, Institute of Animal Science and Veterinary Medicine, Shandong Academy of Agricultural Sciences, Jinan 250100, China; 2Key Laboratory of Livestock and Poultry Multi-Omics of MARA, Jinan 250100, China; 3School of Life Sciences, Shandong Normal University, Jinan 250014, China; 4Shandong Center for Quality Control of Feed and Veterinary Drug, Jinan 250100, China

**Keywords:** *G. parasuis* LPS, inflammation, ATP, TLR4, P2X7R, acute inflammatory response, pharmacological target

## Abstract

*Glaesserella parasuis* (*G. parasuis*), an important pathogenic bacterium, cause Glässer’s disease, and has resulted in tremendous economic losses to the global swine industry. *G. parasuis* infection causes typical acute systemic inflammation. However, the molecular details of how the host modulates the acute inflammatory response induced by *G. parasuis* are largely unknown. In this study, we found that *G. parasuis* LZ and LPS both enhanced the mortality of PAM cells, and at the same time, the level of ATP was enhanced. LPS treatment significantly increased the expressions of IL-1β, P2X7R, NLRP3, NF-κB, p-NF-κB, and GSDMD, leading to pyroptosis. Furthermore, these proteins’ expression was enhanced following extracellular ATP further stimulation. When reduced the production of P2X7R, NF-κB-NLRP3-GSDMS inflammasome signaling pathway was inhibited, and the mortality of cells was reduced. MCC950 treatment repressed the formation of inflammasome and reduced mortality. Further exploration found that the knockdown of TLR4 significantly reduced ATP content and cell mortality, and inhibited the expression of p-NF-κB and NLRP3. These findings suggested upregulation of TLR4-dependent ATP production is critical for *G. parasuis* LPS-mediated inflammation, provided new insights into the molecular pathways underlying the inflammatory response induced by *G. parasuis*, and offered a fresh perspective on therapeutic strategies.

## 1. Introduction

*Glaesserella* (*Haemophilus*) *parasuis* (*G. parasuis*), a gram-negative bacterial species, is the etiologic agent of pigs Glässer’s disease which is characterized by fibrinous polyserositis, polyarthritis and meningitis in pigs [1,2]. In addition, it can be a contributor to swine respiratory disease and is found as a commensal bacterium in the nasal cavity of healthy swine [3]. Recently, *G. parasuis* has become one of the major causes of nursery morbidity and mortality in swine herds, resulting in significant economic losses in the pig industry [4]. So far, 15 serovars of *G. parasuis* have been identified, but >20% of isolates have not been isolated yet [5,6]. The serovar is thought to be an important virulence marker in *G. parasuis* [7]. *G. parasuis* serovars 4, 5, and 13 are the current epidemic strains in China, according to epidemiological studies, with serovar 5 of the organism being considered to be highly virulent and serovar 4 to be moderately virulent. [8,9]. Therefore, managing infection brought on by *G. parasuis* is essential since it is one of the most significant bacterial respiratory infections in pigs. Porcine alveolar macrophages (PAMs) are regarded as a crucial line of defense against *G. parasuis* infection in outbreaks of Glässer’s disease [10]. PAMs release pro-inflammatory and anti-inflammatory cytokines and chemokines to draw leucocytes to the infection site after recognizing the cell structures on the surface of the bacterium, phagocytosing, and lysing it [11,12,13]. However, the factors responsible for systemic infection and inflammatory responses of *G. parasuis* have not yet been fully clarified. Thus, the discovery of novel regulatory factors of *G. parasuis*-induced inflammatory responses may be an alternative strategy for the prevention and control of Glasser’s disease in swine production systems.

Because of sickness, aging, or damage, many cells die at this certain point. Defects can impair cell development and ultimately result in a number of illnesses, such as autoimmune disorders, cancer, or infections [14]. Recently, the field of cell death has rapidly advanced, and multiple cell death pathways have been discovered, including apoptosis, necroptosis, pyroptosis, ferroptosis, and autophagy-dependent cell death. Studies have shown that a large number of effectors of cell death can regulate activation of the NOD-like receptor (NLR) family pyrin domain containing 3 (NLRP3) inflammasome, and NLRP3 inflammasome activation can lead to cell death [15,16]. At the moment, it is widely acknowledged that ligands for Toll-like receptors (TLRs), cytokine receptors (such as the IL-1 receptor and the TNF-α receptor), or NLRs can cause the activation of the transcription factor NF-κB and boost the production of NLRP3 and pro-IL-1β [17,18]. Lipopolysaccharide (LPS) is the most abundant component within the cell wall of Gram-negative bacteria, playing a vital role in the way bacteria interact with the environment and the host. LPS can lead to an acute inflammatory response toward pathogens [19,20]. Toll-like receptor 4 (TLR4), acting as a receptor for LPS, has a pivotal role in the regulation of immune responses to infection [21]. The binding of LPS to TLR4 leads to the activation of NF-κB which plays a crucial role in regulating the transcription of genes related to innate immunity and inflammation responses in the lungs and in monocytes [22].

Trimeric, non-selective cation channels P2X receptors are triggered by extracellular ATP. Because it plays a role in the pathways of apoptosis, inflammation, and tumor growth, the P2X7 receptor subtype is a therapeutic target [23,24]. Acute immobilization stress has been shown to activate P2X7 receptors in a significant quantity of extracellular ATP, which in turn activates NLRP3 and causes the production of inflammatory cytokines [25]. The P2X7R also activates intracellular pathways unrelated to the inflammasomes but frequently associated with them in order to increase inflammation. The activation of NF-κB, a transcription factor that regulates the production of various inflammatory genes such as TNFα, COX-2, and IL-1β, is perhaps one of the best characterized [26,27].

In this research, we explore the role of the ATP/P2X7 receptor axis on *G. parasuis*-induced Glässer’s disease, and the contribution of NLRP3 inflammasome to this pathological process. To further investigate the underlying causative processes of Glässer’s disease, we also explored the effects of various antagonist, agonists, and pathway inhibitors on P2X7 expression and activation. Collectively, these findings could provide a novel viewpoint on treatment options for Glässer’s disease.

## 2. Materials and Methods

### 2.1. Bacterial Strain and Cell Culture

*G. parasuis* serovar 5 stain LZ was isolated in our lab. Bacteria were grown on Trypticase Soy Agar and in Trypticase Soy Broth, respectively (TSA and TSB; OXOID), at 37 °C with the addition of 0.01% nicotinamide adenine dinucleotide (NAD) and 5% (v/v) inactivated bovine serum.

The RPMI1640 medium (Solarbio, Beijing, China) containing 10% fetal bovine serum (FBS) (10091148, Gibco, New Zealand) and 1% pen/strep solution (Solarbio, China) was used to maintain porcine alveolar macrophages (PAM) 3D4/2 cells (ATCC: CRL-2845) at 37 °C in a 5% CO_2_ incubator.

### 2.2. Cell Viability Assay

To determine cell viability, the Cell Counting Kit-8 (CCK-8) assay (Beyotime, Shanghai, China) was used. Briefly, in a 96-well plate, PAM cells were planted and either received *G. parasuis* LZ/LPS treatment or not. After 24 h, 10 μL of CCK-8 solution was added to each well and incubated at 37°C for 2 h. The absorbance at a wavelength of 450 nm was read using a microplate reader (SpectraMax^®^ M5, Molecular Devices, San Jose, CA, USA).

### 2.3. LPS Extraction and Quantification

LPS component of *G. parasuis* LZ was extracted using a Lipopolysaccharide Isolation Kit (Sigma, MAK339, St. Louis, MO, USA). LPS concentrations were determined with Pierce LAL Chromogenic Endotoxin Quantitation Kit (Thermo Fisher Scientific, New York, NY, USA) following the manufacturer’s instructions. In the RPMI1640 medium, LPS was diluted to a storage concentration of 1 mg/mL.

### 2.4. EdU (5-ethynyl-2′-deoxyuridine) Incorporation Assay

The BeyoClick^TM^ EdU Cell Prolifer-ation Kit with Alexa Fluor 555 (Beyotime Biotechnology, Haimen, China) was used to conduct cell proliferation tests in accordance with the manufacturer’s recommendations. PAM cells were treated, then incubated with 10 μm EdU for 2 h at 37 °C. Then cells were subjected to 4% para-formaldehyde fixation and 0.5% Triton X-100 permeabilization steps at room temperature. After the fixatives were removed, 2% BSA in PBS was used to wash the cells. PAM cells were stained with DAPI and treated in Click Additive Solution while being shielded from light. In the following step, a Leica SP8 confocal microscope was used to capture the fluorescence images of the EdU inclusion samples.

### 2.5. ATP Assays

The ATP levels of infected PAM cells were detected by an Enhanced ATP Assay Kit (S0027, Beyotime Biotechnology, Shanghai, China) based on the manufacturer’s instructions. Total ATP levels of PAM cells were quantified by firefly luciferase detection using a luminometer (Tecan Infinite 200pro) and calculated the ATP concentrations (nmol/μg) were based on ATP standard curve.

### 2.6. Enzyme-Linked Immunosorbent Assay (ELISA)

The samples in the medium during cell culture were collected at 4 °C and then added to a 96-well ELISA plate. To measure releases of inflammation-related cytokines from the cells, IL-1β Porcine ELISA Kit (ESIL1B, Invitrogen, Carlsbad, CA, USA) was performed according to the instructions. The absorption value at 450 nm was read by a microplate reader (SpectraMax^®^ M5, Molecular Devices).

### 2.7. RNA Isolation and cDNA Synthesis

24 h after cells were treated with *G. parasuis* LZ, total RNA was extracted using the TRIzol (Life Technologies, Grand Island, NY, USA) technique. After re-suspending whole RNA pellets in RNase-free water, RNA was measured using 260/280 UV spectrophotometry. Next, potentially contaminated DNA was removed by treating the samples with DNase I (Life Technologies). Then, in a 20 μL reaction mixture, 1 μg of total RNA from each sample was reverse transcribed using a ReverTra Ace qPCR RT Kit (TOYOBO, Osaka, Japan) to produce first-strand cDNA. The cDNA was then placed in a freezer before being used.

### 2.8. Quantitative Reverse Transcription Polymerase Chain Reaction (qRT-PCR)

qRT-PCR was performed to measure mRNA expression with the following primers (IL-1β-F: TCTGCCCTGTACCCCAACTG, IL-1β-R: CCCAGGAAGACGGGATTT; β-actin-F: TCTGGCACCACACCTTCT, β-actin-R: GATCTGGGTCATCTTCTCAC). qRT-PCR was performed with SYBR^®^ Green Real-time PCR Master Mix (TOYOBO, Osaka, Japan). cDNA synthesized in 2.7 was used in this chapter. The following cycling circumstances existed: after a denaturation stage at 95 °C for 30 min, 40 cycles of conventional PCR are performed. Melting curve analysis was used to determine the amplified products’ specificity. The 2^−ΔΔCt^ technique was used for quantification. The expression of β-actin mRNA, which was consistent across all samples, was used to standardize gene expression values.

### 2.9. Western Blot

By lysing the cells with ice-cold RIPA buffer supplemented with a protease inhibitor cocktail, total cellular protein lysates were produced (Merck Millipore, Darmstadt, Germany). Following BCA protein quantification, samples were run through SDS-PAGE and then transferred to PVDF membranes. Membranes were incubated with the primary antibodies for an overnight period at 4 °C and with the secondary antibodies for an hour at room temperature following blocking with 5% skim milk. Then, the membrane was visualized with enhanced chemiluminescence and quantified by densitometry. All proteins were normalized to the level of β-actin. The main antibodies were mouse anti-β-actin antibodies and those against NF-κB, p-NF-κB, GSDMD, NLRP3, IL-1β, caspase1, and P2X7 receptor from Cell Signaling Technology in the United States. The secondary antibodies were goat anti-rabbit and goat anti-mouse antibodies (Beyotime, China). Image J software was used to quantify the gray values of protein bands.

### 2.10. Immunofluorescence and Imaging Analysis

PAM were plated on a laser confocal Petri dish. Following the desired treatments, cells were fixed with 4% paraformaldehyde for 10 min and permeabilized with 0.25% Triton X-100 at room temperature for 15 min. Cells were blocked with 5% goat serum for 50 min at room temperature before being incubated with primary NF-κB antibodies (1:200) overnight at 4 °C. The cells were stained with secondary antibodies (1:400) for 1 h after being washed with PBS. All dishes were mounted after being DAPI stained to identify nuclei. All slides were then mounted with ProLongTM Gold Anti-fade mountant. A Leica SP8 confocal microscope was used to capture the immunofluorescence images. 

### 2.11. Plasmids and Transfection

Plasmids, negative control (sense UUCUCCGAACGUGUCACGUTT, antisense ACGUGACACGUUCGGAGAATT) and TLR4-siRNA (sense CAG-GAAUCCUGGUCUAUAATT, antisense UUAUAGACCAGGAUUCCUGTT), are were synthesized by Sangon (China). Lipofectamine^TM^ 3000 (Invitrogen, Carlsbad, CA, USA), transfections were carried out in accordance with the manufacturer’s instructions. In a nutshell, PAM cells were plated in six wells and transfected with 1 mg of plasmid when they were 30–50% confluent. After 24 h of incubation, cells were treated with LPS for further expression.

### 2.12. Plasmids and Transfection

Statistical Analysis: The reported results were statistically evaluated using the paired Student’s t-test method and comparisons between more than two groups were obtained using ANOVA. The reported values are expressed as mean standard errors (SEM). The graphs were plotted using GraphPad Prism version 7.0 (GraphPad Software, La Jolla, CA, USA). Asterisks were used to denote significant values (* *p* < 0.05 and ** *p* < 0.001), whereas *ns* values (*p >* 0.05) were used to denote non-significant values. Each experiment included at least three replicates.

## 3. Results

### 3.1. G. parasuis LPS Enhanced the Mortality and the ATP Level of PAM Cells

We first examined the effect of *G. parasuis* on the viability of PAM cells. PAM cells were treated with *G. parasuis* LZ at MOI = 10 for 8 h. Compared with the mock group, the viability of PAM cells in the *G. parasuis* LZ group was lower (** *p* < 0.01) (Figure 1A). As well, the LPS of *G. parasuis* LZ also resulted in the cell viability decreases when compared with the mock group (** *p* < 0.01) (Figure 1B). To further investigate the effect of *G. parasuis* LZ and LPS on PAM proliferation, EdU staining was utilized. Results of the EdU staining showed that red fluorescence which represents proliferating PAM cells is significantly inhibited by *G. parasuis* LZ and LPS compared with the mock group (** *p* < 0.01) (Figure 1C,D). Extracellular ATP causes the cell membrane to become permeable and induces changes within the cell that could lead to apoptosis [27]. We test the level of extracellular ATP, and found that *G. parasuis* LZ and LPS significantly enhanced ATP levels (** *p* < 0.01) (Figure 1E,F). These results suggested that LPS-enhanced mortality may have a relationship with elevated extracellular ATP levels, and LPS may play a key role in the pathogenesis of *G. parasuis*.

### 3.2. ATP-Induced Pyroptosis and Activated P2X7R Pathway

Although most of the ATP is located intracellularly, it is released into the extracellular space under specific conditions, where it is a relevant signaling molecule. It activates P2X7 and increases inflammatory cytokine levels [28]. So we hypothesized that LPS could induce cellular inflammation by releasing ATP. In order to test it, we regulated the concentration of extracellular ATP in different ways, then observed the effect on IL-1β. The expression of IL-1β in the ATP-added group was higher than *G. parasuis* LZ only group (** *p* < 0.01) (Figure 2A). Nigericin (similar to ATP) also enhanced the expression of IL-1β. While apyrase (a highly active ATP-diphosphohydrolase) reduced the enhanced IL-1β level (** *p* < 0.01) (Figure 2A). As well, in Figure 2B, similar results were shown. We also test the mRNA level of *IL-1β*, and the results were consistent with Figure 2B. As shown in Figure 2D, LPS accelerated the expressions of P2X7R and NLRP3, and Nigericin further increase the expressions (** *p* < 0.01). We also tested the expressions of NF-κB and p-NF-κB, and found that NF-κB was activated by LPS (** *p* < 0.01), and Nigericin enhanced the expression (* *p* < 0.05). These results revealed that LPS-induced release of ATP-activated inflammation.

Physiological roles for GSDMD in both pyroptosis and IL-1β release during inflammasome signaling have been extensively characterized in macrophages and other mononuclear leukocytes. Assembly of N-GSDMD pores in the plasma membrane markedly increases its permeability to macromolecules, metabolites, ions, and major osmolytes, resulting in the rapid collapse of cellular integrity to facilitate pyroptosis [29]. As well, in this study, LPS treatment prominently increased the expression of N-GSDMD (** *p* < 0.01) (Figure 2D), Nigericin further increased the expression of N-GSDMD (* *p* < 0.05) which meant that pyroptosis was activated. All these results suggested that ATP-induced pyroptosis was through ATP/P2X7R pathway.

### 3.3. LPS-Induced Pyroptosis through Activated P2X7R Pathway

To further explore the relationship between P2X7R and pyroptosis, we used 10 μM A740003 (P2X Receptor Antagonist) to treat PAM cells. First, we tested the expression of P2X7R, and found that LPS-enhanced P2X7R was inhabited by A740003. This result meant A740003 worked very well (* *p* < 0.05) (Figure 3A). Then the expression of NLRP3 was observed, A740003 also reduced NLRP3 level significantly (* *p* < 0.05) (Figure 3A), P2X7R was involved in LPS-induced pyroptosis. As well, A740003 inhibited the expression of NF-κB and p-NF-κB compared with cells infected with the LPS group (** *p* < 0.01), meaning that NF-κB may be downstream of P2X7R in this study. When treated with A740003, the level of N-GSDMD was reduced compared with LPS-only group (* *p* < 0.05) (Figure 3B). As well, the level of f IL-1β showed the same result (Figure 3C). We tested A740003 influence on the PAM cells’ survival rate, and found that LPS increases the mortality of PAM cells, when treated with A740003, the mortality decreased (* *p* < 0.05). According to the results of immunofluorescence, NF-κB p65 expression was elevated and more protein entered into the nucleus. These results indicated that the P2X7R pathway plays a central role in the pathogenesis of *G. parasuis*.

### 3.4. NLRP3 Was Involved in the Formation of Inflammation

To better verify the role of the formation of inflammation in cell death, MCC950 (a potent and specific inhibitor of the NLRP3 inflammasome) was utilized in this study. First, we treated cells with different concentrations of MCC950, then observed the expression of NLRP3. Compared with the LPS group, MCC950 markedly reduced the expression of NLRP3 in a concentration-dependent manner (** *p* < 0.01) (Figure 4A). We also detected the expression of caspase 1, showing the same rule (Figure 4A). Subsequently, we tested the level of GSDMD. Compared with the LPS group, MCC950 could significantly reduce the expression of GSDMD (** *p* < 0.01) (Figure 4B). Then we tested the content of IL-1β in the culture medium by ELISA, and found that MCC950 also significantly reduced the secretion of IL-1β (** *p* < 0.01) (Figure 4C). Finally, the cell survival rate was measured by CCK8, and data showed MCC950 could significantly reduce the cell mortality rate that was increased by LPS (** *p* < 0.01). These results suggested that the formation of inflammasome bodies plays a key role in *G. parasuis* infection.

### 3.5. LPS Induced Inflammation in a TLR4-Dependent Manner

Toll-like receptor 4 (TLR4), acting as a receptor for LPS, has a pivotal role in the regulation of immune responses to infection [21]. The binding of LPS to TLR4 leads to the activation of NF-κB which plays a crucial role in regulating the transcription of genes related to innate immunity and inflammation responses in the lungs and in monocytes [22]. To prove that TLR4 plays an important role in *G. parasuis* infection, we used miRNA silencing technology to verify it. First, we tested the silence efficiency of siRNA and found that the siRNA significantly reduced the mRNA level of *TLR4* (** *p* < 0.01), meaning that this siRNA worked well (Figure 5A). Then we observed the effect of TLR4 on ATP levels. Compared with the negative control group, we found that after silencing TLR4, ATP level decreased significantly (** *p* < 0.01) (Figure 5B). In addition, silencing TLR4 significantly restored cell death caused by LPS (** *p* < 0.01) (Figure 5C). Then, we detected the influence of TLR4 on the downstream inflammatory pathway, and found that the expressions of p-NF-κB and NLRP3 decreased, and TLR4 knockout decreased the activation of the NLRP3 inflammasome (** *p* < 0.01) (Figure 5D). These data evidently suggest that LPS induced inflammation in a TLR4-dependent manner. 

## 4. Discussion

*G. parasuis* is the source of Glässer’s disease, which can lead to acute septicemia in non-immune high-health status pigs of all ages and cause instances of arthritis, fibrinous polyserositis, severe pneumonia, and meningitis in piglets worldwide [30]. In this research, we explored the role of the ATP/P2X7 receptor axis on *G. parasuis*-induced Glässer’s disease, and the contribution of NLRP3 inflammasome to this pathological process. 

Bacterial lipopolysaccharides (LPS) are the major outer surface membrane components present in almost all Gram-negative bacteria and act as extremely strong stimulators of innate or natural immunity in diverse eukaryotic species ranging from insects to humans [31,32]. No matter the kind of bacteria involved or the infection location, bacterial adaptation alterations, such as modification of LPS production and structure, are a common motif in infections [33,34]. Generally speaking, these modifications cause the immune system to evade detection, persistent inflammation, and enhanced antimicrobial resistance [35]. LPS derived from *Escherichia coli* (*E. coli*) is a well-characterized inducer of inflammatory response in vivo that activates cytokine expression via NF-κB and MAPK signaling pathway in a TLR4-dependent manner [36]. According to studies, *pseudomonas aeruginosa* (*P. aeruginosa*) LPS changes appear to be a key element in this pathogen’s ability to adapt to chronic infection. Over the duration of the chronic *P. aeruginosa* infection, decreased LPS immunostimulatory potential helps the immune system avoid detection and survive [37]. It has been reported that anti-LPS antibodies can protect against mortality caused by hematogenous *Haemophilus influenzae* type b meningitis infections in infant rats [38]. In this study, we found that *G. parasuis* LZ induced cells death and severe inflammation in PAM cells (Figure 1A and Figure 3A), and LPS derived from *G. parasuis* LZ treatment group also has similar phenomena, these suggested that *G. parasuis* LPS plays a key role in host-pathogen interactions with the innate immune system.

Pyroptosis is an inflammatory form of cell death that is brought on by certain inflammasomes [39,40]. This kind of cell death causes the cleavage of gasdermin D (GSDMD) and the activation of dormant cytokines like IL-18 and IL-1β. Cell enlargement, lysis of the plasma membrane, fragmentation of the chromatin, and release of the pro-inflammatory substances inside the cell are all effects of pyroptosis [41]. The conventional inflammasome pathway, a noncanonical inflammasome pathway, and a newly discovered pathway are the pathways that cause pyroptosis [42,43]. Caspase-11 may selectively attach to the lipid A of intracellular LPS, which causes it to oligomerize, engage its proteolytic activity, and cleave the GSDMD to create a large number of holes in the cell membrane, ultimately causing membrane lysis and pyroptosis [44]. As well, the extracellular LPS stimulation of neutrophils can also activate the TLR4-P38-Cx43 pathway to autocrine ATP extracellularly [45]. The extracellular ATP could gather NLRP3 inflammasomes and subsequently activate the pro-caspase 1 through the P2X7 pathway, resulting in pyroptosis [46]. In this study, we found that *G. parasuis* LZ LPS induced cell death and promoted the increase of ATP content, thus activating the P2X7 pathway, promoting the development of IL-1β, and cleavage of GSDMD, leading to pyroptosis. This is consistent with the canonical inflammasome pathway. Luo et al. have reported that *G. parasuis* induces an inflammatory response in PAM cells through the activation of the NLRP3 inflammasome signaling pathway [30], which is consistent with our result.

*G. parasuis*, an opportunistic pathogen of the lower respiratory tract of pigs, is also associated with pneumonia and is involved in the porcine respiratory disease complex [47]. Secondary *G. parasuis* infection enhances highly pathogenic porcine reproductive and respiratory syndrome virus (HP-PRRSV) infection-mediated inflammatory responses [48]. The polarization of LPS-stimulated PAMs toward M1 PAMs greatly reduces PRRSV replication [49], mainly because LPS reduced the level of CD163 expression to inhibit PRRSV infection via TLR4-NF-κB pathway [30]. In this study *G. parasuis* LPS activated inflammatory responses through TLR4-NF-κB pathway, and combined with the above reference, we got the hypothesis that *G. parasuis* infection can significantly inhibit PRRSV replication through downregulation of CD163 expression via TLR4-NF-κB pathway. However, this hypothesis needs further verification.

In conclusion, *G. parasuis* induced PAM cell damage mainly through included pro-inflammatory and pro-pyroptosis events. The NLRP3 inflammasome in PAM cells plays a crucial role in *G. parasuis*-induced cells death and both TLR4- and P2X7R-dependent pathways are alternative signaling pathways required for NLRP3 inflammasome activation during the development of *G. parasuis*-induced Glässer’s disease. This work provides new insights into the molecular pathways underlying the inflammatory response induced by *G. parasuis* and a new perspective to inform the targeted treatment of *G. parasuis*-induced Glässer’s disease.

## Figures and Tables

**Figure 1 cells-12-00751-f001:**
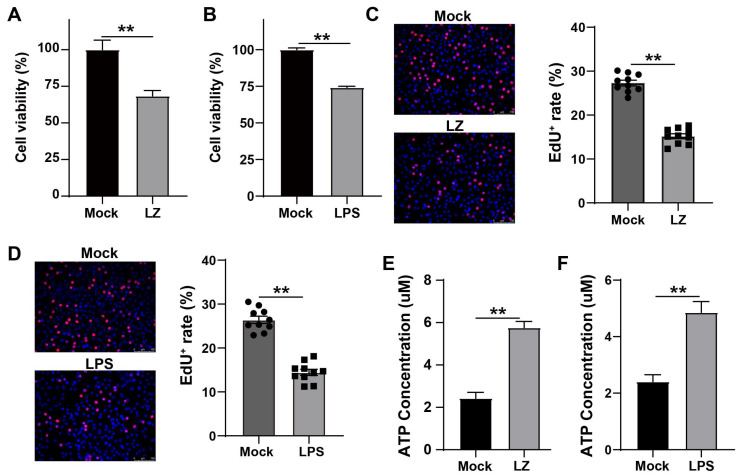
*G. parasuis* LPS enhanced the mortality and the ATP level of PAM cells. (**A**,**B**) Quantification of mortality. PAM cells were treated with *G. parasuis* LZ (**A**) and 50 μg/mL LPS (**B**) for 8 h, and cell viability was measured by CCK-8. (**C**,**D**) Representative images of cell proliferation were determined by EdU cell proliferation assay, and quantification of EdU^+^ cell. *n* = 10. (**E**,**F**) ATP levels in PAM cells. After PAM cells were affected with *G. parasuis* at MOI = 10 for 8 h, PAM cells were lysed and the cell lysates were analyzed for ATP levels. Data represent mean ± SEM, *n* = 3, ** *p* < 0.01.

**Figure 2 cells-12-00751-f002:**
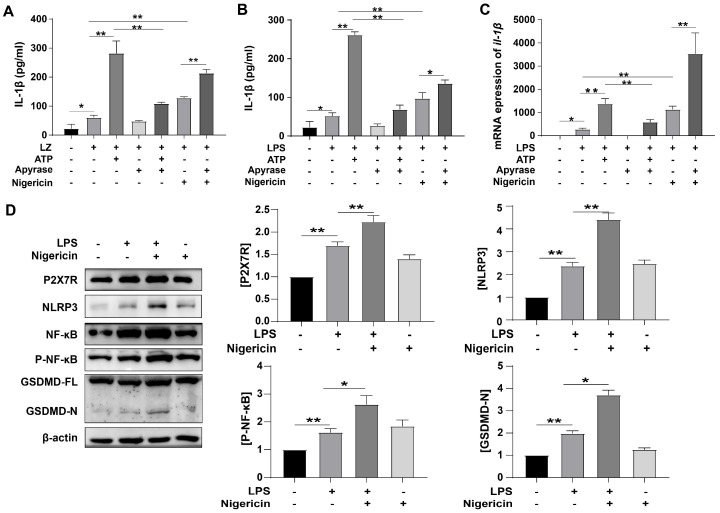
ATP induces inflammation and increases P2X7 expression. (**A**,**B**) ELISA analysis of IL-1β normalized to the control. PAM cells were treated with *G. parasuis* LZ (**A**) /50 μg/mL LPS (**B**) in the presence and absence of ATP, apyrase, and nigericin. (**C**) mRNA expression measured by qRT-PCR for *IL-1β* level normalized to the control. (**D**) Western blot analysis of P2X7R, NLRP3, NF-κB, p-NF-κB, and GSDMD expression in PAM cells. Cells were treated with or without LPS in the presence and absence of nigericin. All proteins were normalized to the level of β-actin. Data represent mean ± SEM, *n* = 3, * *p* < 0.05, ** *p* < 0.01.

**Figure 3 cells-12-00751-f003:**
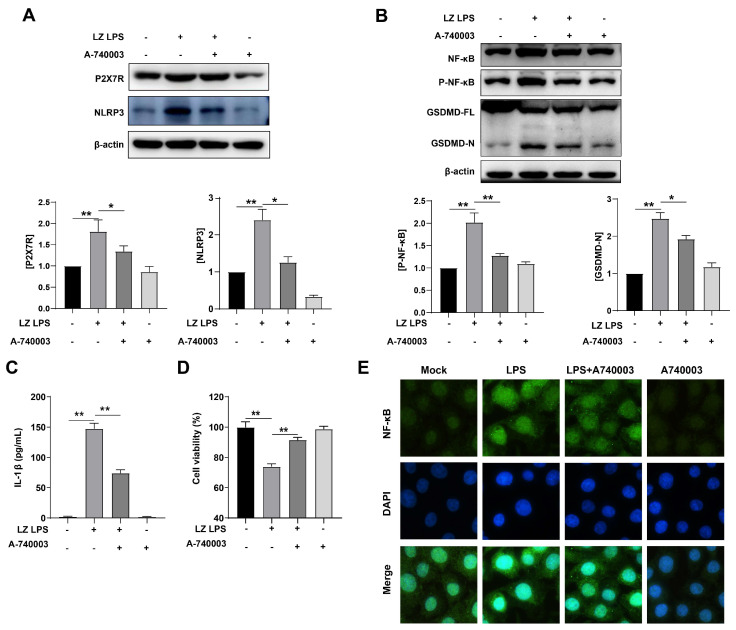
A740003 regulates P2X7 function and inhibits inflammation. (**A**) Western blot analysis of P2X7R, NLRP3 expression in PAM cells. All proteins were normalized to the level of β-actin. Cells were treated with or without 50 μg/mL LPS in the presence and absence of A-740003 (10 μM) (**B**) Western blot analysis of NF-κB, p- NF-κB, and GSDMD expression in PAM cells. All proteins were normalized to the level of β-actin. (**C**) ELISA analysis of IL-1β normalized to the control. (**D**) Quantification of mortality. After PAM cells were treated, cell viability was measured by CCK-8. (**E**) Representative images of immunofluorescence staining. Differentiated PAM cells were treated with or without LPS in the presence and absence of 0.1 μM A-740003. Nuclei were stained by DAPI in Blue and NF-kB p65 were stained in green, then observed using an inverted fluorescence microscope, 100×. Data represent mean ± SEM, *n* = 3, * *p* < 0.05, ** *p* < 0.01.

**Figure 4 cells-12-00751-f004:**
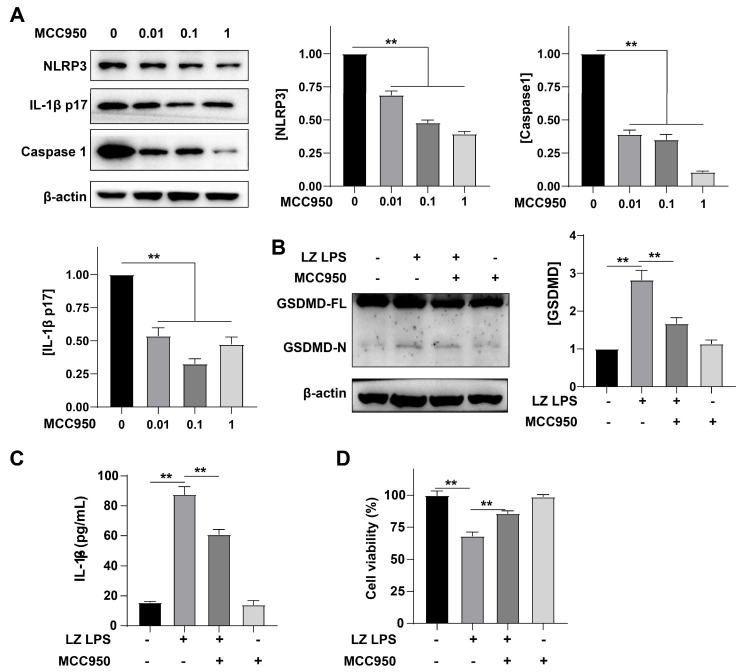
MCC950 reduced NLRP3 expression and inhibited inflammation. (**A**) Western blot analysis of NLRP3, IL-1β p17, and cleaved caspase1 expression in PAM cells. PAM cells were treated with 50 μg/mL LPS in the presence of a different concentration of MCC950 (0, 0.01, 0.1, 1 μM). All proteins were normalized to the level of β-actin. (**B**) Western blot analysis of GSDMD expression in PAM cells. PAM cells were treated with LPS in the presence of 0.1 μM MCC950. (**C**) ELISA analysis of IL-1β normalized to the control. (**D**) Quantification of mortality. After PAM cells being treated, cell viability was measured by CCK-8. Data represent mean ± SEM, *n* = 3, ** *p* < 0.01.

**Figure 5 cells-12-00751-f005:**
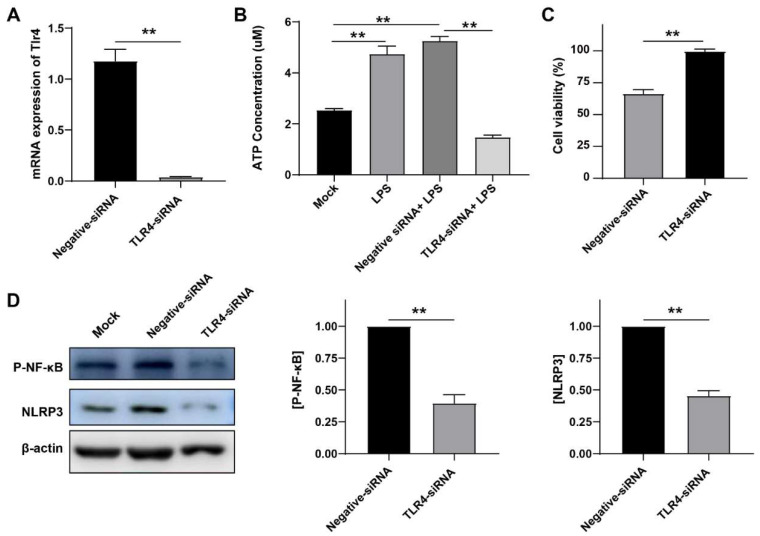
Inhibition of TLR4 reduced inflammation and increased cell viability. (**A**) mRNA expression measured by qRT-PCR for Tlr4 level normalized to the negative control. (**B**) ATP levels in PAM cells. After cell transfection, and cells were treated with LPS. PAM cells were lysed and the cell lysates were analyzed for ATP levels. (**C**) Quantification of mortality. After cell transfection, all PAM cells were treated with 50 μg/mL LPS, and test the cell survival rate by CCK8. (**D**) Representative Western blot analysis of NLRP3 andIL-1β p17 were normalized based on the internal control β-actin. After cell transfection, all PAM cells were treated with LPS, and tested the expression of NLRP3 and IL-1β p17. Data represent mean ± SEM, *n* = 3, ** *p* < 0.01.

## Data Availability

No large-scale datasets have been generated. Raw data of the experiments and/or materials can be provided upon reasonable request.

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
