# Peer review of "Upregulation of TLR4-Dependent ATP Production Is Critical for Glaesserella parasuis LPS-Mediated Inflammation"

_cells, 2023, doi:10.3390/cells12050751_

Round 1

Reviewer 1 Report

The aim of this work is to contribute in the knowledge of the mechanisms involved in the inflammatory reaction induced by the Glaesserella parasuis (G.parasuis), which is an important pathogenic bacterium for pigs. The infection caused by this bacterium has a relevant impact for the global swine industry. Therefore, firstly the authors show an overview of the relevance to study the G.parasuis infection and the role of the LPS as a major component involved in the inflammation and the disease. In this sense, the authors evaluated the effect of LPS to mediate the activation of porcine alveolar macrophages and the participation of ATP, NLRP3 and P2X7R in this process. The experiments were very well conducted. The results are consistent showing the axis of ATP/P2X7R to induce NLRP3 activation with IL-1beta production followed pyroptosis in response to G.parasuis infection. Despite that, some points should be better discussed and the manuscript revised regarding the grammatical errors for to improve the impact of the paper.

The authors should describe the methodology and the reference of obtaining the alveolar macrophages used in the experiments, the details of cell cultures and the in vitro stimulation process.

Concerning about the proliferation experiment, why the author did not measure the proliferative index calculated?

Reviewer 2 Report

The manuscript entitled “Upregulation of TLR4-dependent ATP production is critical for Glaesserella parasuis LPS-mediated inflammation” is well organized and well prepared. This paper illustrates an interesting story, ATP/P2X7R pathway plays an important role in the cellular inflammatory response induced by LPS of Glaesserella parasuis through TLR4. My suggestion is that only minor changes may be requested before publication. 

1. In this paper, author did not tell us what the treatment concentration of G. parasuis LPS is?

2. In this study, MCC950 and A740003 were used, but the author did not tell us the concentration. Please add the concentration.

3. The author need to define the abbreviation the first time you use it in each part.

4. Line 93: Pig alveolar macrophages (PAM) is primary or PAM3D4/2? Please describe clearly.

5. Line 96: correct the sentence, kindly replace “37C” with “37°C”.

6. Line 128: correct the sentence, kindly replace “IL-1 beta” with “IL-1β” throughout the manuscript.

7. Line 233-234: “All these suggested that ATP induced pyroptosis not only through P2X7R pathway.” Correct this sentence to make it more clearly.

8. Line 298: kindly replace “Tlr4” with “TLR4”.

Reviewer 3 Report

This article describes important and interesting issue dealing with new insights into the molecular pathways underlying the inflammatory response induced with G. parasuis, important pathogenic bacterium which causes tremendous economic losses to the global swine industry. Described findings can be used in pharmacological industry and clinical research. However, I do suggest certain things, which need attention, improvement and clarification to support and strengthen the overall impact of the article.

Points for attention:

Abstract: Maybe to point out that the findings offer a fresh perspective on therapeutic strategies.

Keywords: To increase article’s searchability it would be good to add more keywords, P2X7R, acute inflammatory response, pharmacological target.

Introduction:

Lines 40-41: It would be good to indicate that the serovars 4, 5 and 13 are the current epidemic strains in China.

It is explained in some other papers that the amplified inflammatory cytokine production by macrophages can be result of the synergic action between some viruses (e.g. PRRSV) and LPS. Could authors maybe comment? Is there a possibility in the future to test pigs with Glässer’s disease also on viral pathogens?

Line 43 porcine not prcine

Materials and Methods: 2.8. Quantitative reverse transcription polymerase chain reaction

It is not clearly explained what was the starting material (isolated RNA or synthetized cDNA). In the title is written that the authors performed the quantitative reverse transcription polymerase chain reaction, but when the qRT-PCR cycling conditions were described, the reverse transcription step is missing. Please explain.

Results:

It would be good if the authors would mention results of the statistical analysis in every subchapter (not only in the figure legend).

3.1. G. parasuis LPS enhanced the mortality and the ATP level of PAM cells: line 196- is the number 27 after the word apoptosis cited reference?

Discussion:

Line 312. G. parasuis also causes frequent symptoms of pneumonia.

It would be good if the authors could discuss the results of the statistical analysis in this chapter.

Part with lines 318-360 is too long.

Needs improvements in addition of significance impact of the study.  

Round 2

Reviewer 1 Report

Concerning about the relevant pathogenic action of Glaesserella parasuis (G.parasuis) for pigs and consequent global swine industry. In this manuscript, the authors show the effect of LPS to mediate the activation of porcine alveolar macrophages and the participation of ATP, NLRP3, NF-κB, GSDMD and P2X7R in this process contributing for the knowledge about the mechanisms involved in the immune response for this pathogenic agent. The revised version includes suggested changes and, therefore the manuscript was improved for the publication.
